# Rationale, development and feasibility of group antenatal care for immigrant women in Sweden: a study protocol for the Hooyo Project

Ulrika Byrskog,[1] Malin Ahrne,[2] Rhonda Small,[3] Ewa Andersson,[2] Birgitta Essen,[4] Aisha Adan,[2] Fardosa Hassen Ahmed,[2] Karin Tesser,[5] Yvonne Lidén,[6] Monika Israelsson,[6] Anna Åhman-Berndtsson,[5] Erica Schytt[7,8]

For numbered affiliations see end of article.

**Correspondence to**
Dr Ulrika Byrskog; uby@du.se

## ABSTRACT

**Introduction** Somali-born women comprise a large group of immigrant women of childbearing age in Sweden, with increased risks for perinatal morbidity and mortality and poor experiences of care, despite the goal of providing equitable healthcare for the entire population. Rethinking how care is provided may help to improve outcomes.

**Overall aim** To develop and test the acceptability, feasibility and immediate impacts of group antenatal care for Somali-born immigrant women, in an effort to improve experiences of antenatal care, knowledge about childbearing and the Swedish healthcare system, emotional well-being and ultimately, pregnancy outcomes. This protocol describes the rationale, planning and development of the study.

**Methods and analysis** An intervention development and feasibility study. Phase I includes needs assessment and development of contextual understanding using focus group discussions. In phase II, the intervention and evaluation tools, based on core values for quality care and person-centred care, are developed. Phase III includes the historically controlled evaluation in which relevant outcome measures are compared for women receiving individual care (2016–2018) and women receiving group antenatal care (2018–2019): care satisfaction (Migrant Friendly Maternity Care Questionnaire), emotional well-being (Edinburgh Postnatal Depression Scale), social support, childbirth fear, knowledge of Swedish maternity care, delivery outcomes. Phase IV includes the process evaluation, investigate process, feasibility and mechanisms of impact using field notes, observations, interviews and questionnaires. All phases are conducted in collaboration with a stakeholder reference group.

**Ethics and dissemination** The study is approved by the Regional Ethical Review Board, Stockholm, Sweden. Participants receive information about the study and their right to decline/withdraw without consequences. Consent is given prior to enrolment. Findings will be disseminated at antenatal care units, national/international conferences, through publications in peer-reviewed journals, seminars involving stakeholders, practitioners, community and via the project website. Participating women will receive a summary of results in their language.

### Strengths and limitations of this study

► The feasibility design of the study together with the nested process evaluation will contribute valuable information for future controlled studies and in the design of antenatal care interventions that target inequalities in health between immigrants and non-immigrants.

► By emphasising dialogue, a person-centred approach and active participation of parents, midwives and bilingual research assistants throughout development and implementation, recruitment and retention are optimised and a relevant, pragmatic and acceptable model of care is likely to result.

► Funding has limited the scope of the study to two sites, with a focus on one group of immigrant women.

## INTRODUCTION

Sweden has a clearly stated goal of providing equitable healthcare for the whole population, which in recent decades has become increasingly diverse. Despite this, studies indicate that pregnancy outcomes among immigrant women are suboptimal compared with those of Swedish-born women.[1] These health disparities point to the need for implementing and evaluating interventions to improve care for immigrant women and their families. This study protocol describes the development of an intervention to improve antenatal care (ANC) for Somali-born women and families giving birth in Sweden.

Somali-born women constitute one of the largest groups of immigrant women of childbearing age in Sweden[2] after more than two decades of political instability in Somalia. Of all immigrant women, they are known to be at high risk of maternal and perinatal morbidity and mortality both in Sweden[3–5] and internationally.[6] The reasons are complex and

include both premigration and postmigration factors. Insufficient healthcare provision and low socioeconomic conditions in the country of origin influence women's health status.[7] After migration to Sweden, later and lower attendance for antenatal care[8] and poorer experiences of maternity care[9] have been reported. Language difficulties and lack of familiarity with care systems and discrimination and suboptimal care are contributing factors.[3 10 11] Furthermore, higher rates of anaemia, insufficient weight gain and infants born small for gestational age[8] indicate that Somali women are not sufficiently reached by advice about prevention or treatment. Lower attendance at childbirth and parent education classes among immigrant women has also been found, with classes most often held in Swedish.[12]

Few measures have been taken in Sweden to reduce barriers to maternity care for immigrant women or to better meet their care needs. Furthermore, few studies have investigated how care might be designed in ways that are both attractive and acceptable to immigrant women and to midwives who are the main providers of care during pregnancy and childbirth. Respect, communication, community knowledge and care tailored to women's circumstances and needs have been proposed as central components in quality maternity care.[13] In line with this, a systematic review in five countries highlighted that immigrant women want the same things from care as non-immigrant women: care should be of high quality and safe, with adequate information and support and attentive to individual needs. The review also found however, that immigrant women consistently reported experiencing additional challenges, including communication difficulties, lack of familiarity with healthcare systems and discrimination.[10]

Language barriers, marginalisation and prevailing cultural stereotypes pose challenges in designing, implementing and evaluating innovative care models including, rather than excluding, immigrants. These factors need to be carefully addressed before widespread change of new models of care are introduced. Participatory approaches have shown promising results[14–16] and reduce the risk of enhancing marginalisation. Focus on communication and meaningful dialogue throughout intervention design, recruitment, data collection, implementation and evaluation is valuable.[15]

In Sweden, the care of pregnant women during normal pregnancies is provided at community-based ANC clinics by registered midwives, based on national and regional guidelines.[17] Included are 8–10 pregnancy check-ups free of charge, traditionally offered through individual appointments (30 min) with the same midwife throughout the pregnancy. Extra visits are booked and specialised medical referrals are made when needed. Parent education including preparation for childbirth and parenting is provided during the individual appointments or in groups/classes, mainly in Swedish and often focusing on first-time parents. Partners are encouraged to participate.

One innovative approach for improving antenatal care is group antenatal care (gANC). This care model incorporates pregnancy assessments and parent education in small groups, either with shorter individual appointments with the midwife at the beginning or end of the group session, or with individual clinical checks by the midwife provided in the group space,[18–20] while the rest of the group can socialise. Self-assessments of blood pressure and urine have also been included in some gANC models.[18 20] Results from studies in the USA, Iran and Sweden report improved satisfaction with care,[19 21–24] improved iron taking, higher birth weight, lower caesarean section rates, more breastfeeding and earlier diagnosis of complications.[23] Higher self-esteem and lower levels of stress, social conflicts and depression have been reported, as have more visits to ANC by fathers-to-be, increased support in social contacts with other parents and reduced costs,[18 19 21 23 25–27] despite more time spent with the midwife throughout pregnancy. gANC thus appears to enhance women's agency, dialogue, time with the midwife and social interaction, and to improve women's views of their care and also some of their pregnancy outcomes. gANC may be especially valuable for specific groups of women, as seen for example in studies among young African-American women in the USA[21 28] and among Karen refugee women in Australia.[15]

The Hooyo Project ('mother' in Somali) is a gANC initiative developed in Sweden in response to the evidence about lack of equity in pregnancy care provision and poorer outcomes for Somali-born women. gANC has not been evaluated among immigrant women and families in Sweden, yet it may well be a model of care that can break down cultural and care barriers, enable better dialogue and understanding between caregivers and immigrant women and improve women's satisfaction with their care, especially if language support is provided.[15]

The overall aim for this study is to develop and test the acceptability and immediate impacts of gANC for Somali women residing in Sweden in an effort to improve experiences of antenatal care, knowledge about childbearing and the Swedish healthcare system, emotional well-being and ultimately, pregnancy outcomes.
Specific objectives include:
1. To establish with Somali-born women and antenatal caregivers the acceptability and feasibility of gANC and to develop an appropriate model to improve outcomes (phases I and II).
2. To establish appropriate recruitment and data collection procedures and outcome measures in consultation with the Somali community and with care providers to evaluate the new model (phase II).
3. To implement and evaluate an agreed model of gANC in partnership with the Somali community and antenatal caregivers (phases III and IV).

## METHODS AND ANALYSIS

### Design

An intervention development and feasibility study including focus group discussions, a historically controlled evaluation and process evaluation.[29 30] The intervention is implemented in two antenatal care clinics in mid-Sweden. The feasibility and piloting includes testing procedures for acceptability, estimating the likely rates of recruitment and retention of women and the calculation of appropriate sample sizes for future controlled trials. Complex interventions that are adjusted to local, specific[31] or personal needs[32] are likely to be more effective than completely standardised models.[29] Key principles underpinning the study are therefore:

► Active involvement of Somali parents and midwives in needs assessment and care design.
► Attention to language and contextual factors.
► Flexibility in study methods to respond to issues as they arise.
► A care model ready to continue or be replicated after the project concludes with only minor adjustments.

Hooyo consists of four phases with process evaluation activities informing study progress. Phase I is the preparation phase, including needs assessment, the development of contextual understanding and building a logic model for the project. Phase II includes the development of the intervention and of the evaluation tools, and phase III involves implementation and evaluation of the intervention using historical controls. In phase IV, the implementation process, the feasibility and the mechanisms of impact are evaluated, including contextual factors, using Moore's Process Evaluation of Complex Interventions as an overarching framework.[30] Phases I and II were conducted in 2016–2017. Recruitment and data collection for the historically controlled evaluation (phase III) and process evaluation (phase IV) commenced in late 2016 and will be completed during 2020. Women in the control group were recruited between October 2016 and April 2018 and recruitment of women into the intervention began in May 2018 and will be complete in December 2019. Final data collection will be completed by July 2020. The paper covers methodological aspects of all four phases of the study.

### Phase I: preparation phase

The purpose of phase I was to understand how current individual care was delivered and experienced and what changes women, partners and midwives thought might improve care, taking into account contextual factors that might have an impact on the development, structure and content of a new model. The aim was also to assess whether a language-supported group antenatal care model might be appropriate and acceptable to address concerns raised about current care.

### Initial dialogue and choosing study settings

In the municipality chosen for site 1, the number of Somali immigrants had increased fivefold during the last decade and the ANC midwives had begun to rethink how they were providing care. At the same time, there was awareness of lower ANC attendance and adverse outcomes among childbearing Somali-born women in Sweden,[8] and a need for more appropriate explanations of ANC routines had been highlighted in interviews with Somali-born women.[33] This prompted a dialogue between the principal investigator and the ANC midwives, outlining the possibility of evaluating an alternative model of care. The ANC clinic is located in a primary healthcare centre staffed by 10 midwives, within a public hospital located outside the city centre. It caters for approximately 75% of all pregnant women in the municipality and almost all of the pregnant immigrant women in the municipality, which in total has 50 000 residents. The majority of the Somali-born women visiting the clinic live in two areas situated approximately five kilometres away, necessitating use of a car or public transport.

At site 2, the ANC clinic is located in a family health centre in a suburb of the capital city. Close collaboration takes place between ANC midwives, social workers, child health nurses and the open playgroup located in the same building. Three ANC midwives are employed, and the target area includes two residential areas with different sociodemographics; with primarily Swedish-born residents in one area and non-Swedish born residents and second-generation immigrants in the other. Somali-born residents have lived in this area for more than 25 years, and midwives reported that most Somali-born pregnant women and families were able to communicate in Swedish. Integration of families of different cultural backgrounds is actively encouraged at the clinic. Initial discussions revealed that the midwives believed group antenatal care for women of mixed cultural backgrounds (with interpreters available if needed) would contribute positively to integration. As a result, it was agreed that integrated groups would be more appropriate and acceptable at this site.

### Mechanisms for engagement: reference group and bicultural research assistants

A project reference group was established comprising research team members, antenatal care midwives and representatives of the Somali community from both sites. Terms of reference were developed, including provision of advice to the research team on design of the intervention, appropriate recruitment processes and data collection methods; development of study questionnaires; networking with Somali associations and ANC clinics as well as contributing to the interpretation and dissemination of findings throughout the project period.

A bilingual Somali research assistant with a healthcare background was employed full time in the project at site one. This enabled networking with Somali community members, bridging language gaps, input into

questionnaire design, recruitment of participants, data collection and arranging and interpreting focus group discussions together with research team members. At site 2, focus group discussions were facilitated by two Somali community workers, including recruitment and interpreting.

### Participatory focus group discussions with Somali parents and antenatal care midwives

To develop understanding of current antenatal care and how it was experienced, phase I included focus group discussions (FGDs) with Somali-born parents and with ANC midwives; presented in brief below and in Ahrne *et al*.[34]

Parents were recruited using purposeful sampling. In total, 16 mothers and 13 fathers with varied length of stay in Sweden and recent experience (<2 years) of ANC were included. Seven ANC midwives were recruited purposefully from both sites and from an additional site where the midwives had previous experience of group antenatal care with Somali-born parents.

Three focus group discussions were held with Somali-born mothers, two with Somali-born fathers and three with ANC midwives. Emerging themes were highlighted by the researchers and crosschecked during the discussions with participants for accuracy.[35 36] Thematic analysis as described by Attride-Sterling[37] was used to interpret data from the FGDs.

### Key findings informing intervention development

Challenges at the system level and in the care encounter in striving for optimal ANC were identified[34] alongside findings that could directly inform the intervention development. Instead of focusing on specific cultural or ethnic aspects of care, the results indicated a need to maintain focus on diversity and individual needs. The need for forums for social interaction, and for dialogue which could bridge gaps between midwives' and women's understanding of the purpose of antenatal care, improve communication and reduce the risks for stereotyping was apparent. Responding to a wide range of health literacy needs, welcoming fathers and safeguarding an open atmosphere were other issues raised. Furthermore, practical issues such as optimal location and times for ANC visits were discussed. The findings contributed valuable input to the programme theory and to the decisions regarding underpinning central principles for the coming intervention (table 1) as well as for the development of the structure and content of the intervention in phase II.

### Phase II: development of the intervention and study evaluation tools

#### Developing group antenatal care

Prior to the study commencing, a review of the literature on migrant women's birth outcomes, experiences of antenatal care and on alternative models of care for migrant women had been undertaken, including group antenatal care. This, together with the experiences of team member

EA in evaluating group antenatal care in Sweden,[19 24 27] led us to investigate whether group antenatal care could be appropriately developed to improve outcomes for Somali-born women. The preliminary findings from the FGDs confirmed language supported gANC as a possible alternative care model, and this was discussed in the reference group. Research team members participated in discussions at each study site during these processes. The intervention development was characterised by constant modifications to fit local prerequisites at both sites, including local guidelines and routines in the ANC clinics, and needs among the women; not all of which were apparent from the start.

#### Underpinning principles

Central findings in the FGDs were the desire for individualised care, a need for forums for dialogue to address variations in health literacy and healthcare understanding and at the same time respond to the request to move beyond stereotypes based on ethnicity or 'culture'. This led us to focus on person-centred care (PCC)[38] as a foundation for the intervention, and on how this could be strengthened and encouraged through active dialogue in group-based care. This was also in line with the ongoing implementation of PCC generally in the health system in Sweden. In searching for a method to support this approach, the midwives proposed motivational interviewing (MI). They had already received some training in MI for use in individual appointments, and now MI for groups was added. Principles of PPC and MI encourage understanding the person as an individual and developing partnership and promoting self-efficacy for which an active and open dialogue is central.[38–40]

#### Addressing language and integration

To respond to the diverse needs identified at the two ANC clinics and to assess advantages and disadvantages with both homogeneous and heterogeneous groups regarding language and cultural background, it was decided that gANC would be offered to Somali-born women at site 1. A large number of Somali migrants had settled in this municipality in recent years, and this meant that many were not yet fluent in Swedish. At site 2, gANC would be offered to all women attending the clinic, in integrated groups. At site 1, the need for a Somali interpreter at each session was anticipated. Since a finding in the FGDs was that continuity of known Somali interpreters had been a previous success factor, it was decided to engage two experienced interpreters who could alternate. At site 2, it was anticipated that the need for interpreters in different languages would vary and thus they would be engaged when needed.

#### The intervention: gANC

The intervention consists of gANC offered at both sites, modified to the needs in each site and group. Women are allocated to a group with women at similar stages of pregnancy within a 4-week gestational age span. From

**Table 1** Logic model of Hooyo including problem statements, conceptual framework and rationale, the Hooyo group antenatal care model, hypothesised mechanisms of effect and desired outcomes

| Problem statement | Conceptual framework and rationale | Hooyo: a group approach to improving ANC | Hypothesised mechanisms of effect | Desired outcomes |
|---|---|---|---|---|
| Current ANC in Sweden may not provide equitable care for Somali-born women: <br>▲ Lower participation in antenatal care <br>▲ Poorer birth outcomes <br>▲ Communication difficulties <br>▲ Lack of familiarity with Swedish antenatal care structures <br>▲ Lower attendance in parental education <br>▲ Negative attitudes and suboptimal care <br><br>Initial FGDs with Somali-born parents/ANC midwives highlight need for: <br>▲ Improved communication and dialogue <br>▲ Bridging gaps between divergent health literacy knowledge <br>▲ Care free from generalisations, tailored to individual needs <br>▲ Clearly described expectations regarding partner's role | Core values for quality care: respect, communication, community knowledge and understanding <br>▲ Person-centred care to identify and address women's individual needs <br>▲ Continuity of care for positive care experiences and health outcomes <br><br>Group antenatal care a promising alternative to individual visits: <br>▲ More positive views of care <br>▲ Some positive impacts on birth outcomes <br>▲ More time with midwives and more comprehensive parental education <br>▲ In Sweden studied with Swedish-speaking groups only <br><br>Key underpinning principles: <br>▲ Active involvement of Somali parents/ midwives in assessment and care design <br>▲ Attention to language and contextual factors <br>▲ Flexibility in study methods to respond to issues as they arise <br>▲ A care model ready to continue or replicate after project ending with minor adjustments | Language supported group antenatal care involving <br>▲ 8–9 group sessions 1 1/2 hours with 6–8 women (partners welcome) from gest. week 24 <br>▲ Facilitated by two midwives assisted by interpreter <br>▲ Brief individual midwife check-ups incorporated <br>▲ Childbirth/parenting themes with focus on dialogue and discussion <br>▲ Motivational interviewing for groups as a vehicle for focusing care on women's needs <br>▲ Adjustments based on local needs at each site: Site 1: Groups specifically for Somali-born Site 2: Groups with diverse backgrounds and languages | Interpreter-supported group dialogue facilitated by midwives will result in <br>▲ Improved communication →better suited care <br>▲ More time for discussions →mutual understandings in views around childbirth and health promotion → strategies for improving outcomes <br>▲ An additional arena for social contact and support →increased well-being <br>▲ Combining pregnancy check-ups with groups →motivation for attending ANC, and parental education <br>▲ Common language/ background → understanding and empower women to raise voices in having needs addressed <br>▲ Mixed groups →integration and understanding through cross-language/ culture interactions | Women: <br>▲ Happier with the ANC <br>▲ More confident in and knowledgeable about the pregnancies <br>▲ Improved well-being <br>▲ Improved attendance at antenatal care visits <br>▲ Improved uptake of health advice <br>▲ Ultimately improved pregnancy outcomes <br>Partners: <br>▲ Feeling welcomed and included <br>▲ Increased understanding of expectations <br>Midwives: <br>▲ Improved understanding of womens needs <br>▲ Feel better able to share health knowledge in meaningful ways <br>▲ Provide more supportive, non-judgemental care <br>▲ Positive about benefits of group care |

ANC, antenatal care; FGD, focus group discussion.

the second visit (gestational weeks 20–26), they receive gANC. Each group will consist of six to eight women and partners (or another support person) for optimal group dynamics and dialogue. While reports of gANC internationally involve numbers in groups of 5–20 in low-income to middle-income countries[41] and 8–12 in high-income countries,[42] the need to provide interpreting during groups sessions led to the choice of a somewhat smaller group size as being appropriate to allow all group members a chance to participate in the discussions. Partners are generally encouraged to participate in Swedish ANC, and hence this will be the point of departure also for the gANC but with the final decision about partner participation taken by the women included in each group. Frequency and number of sessions follow the national Swedish programme for antenatal care, that is, eight to nine appointments during a normal pregnancy. Each visit includes a group session for 1 hour, facilitated by one of two midwives assigned to the group and with interpreter assistance. Although each group session has a focus on a theme related to pregnancy, birth or parenting in line with national recommendations,[17] particular emphasis will be given to issues and questions raised by the participants. Group care allows women to talk with each other and share their experiences in their own language. They also hear midwives' responses to other women's questions, expanding the dialogue and information sharing that can occur between midwives and women. The presence of an interpreter facilitates communication between the women and the midwives.

Alongside the group discussion, each woman also has a 15 min individual appointment with the midwife responsible for her care, who is the same midwife for each appointment according to Swedish National Guidelines.[17] Routine pregnancy controls are carried out, and the woman can also raise any particular issues with her midwife during this time. In total, the time for each visit will be approximately 75 min instead of 30 min, as is common for standard individual care (though this does not take into account the time midwives also spend providing childbirth and parenting classes). Additional individual appointments can be booked as needed if medical or other issues arise. This model for group antenatal care draws on the model used in a previous Swedish study by Andersson *et al*[19 24 27] but differs in that two midwives and a language interpreter are present throughout the group sessions, and the principles of PCC and MI are explicitly identified as the theoretical underpinnings for care in the group.

### Development of evaluation tools
#### Questionnaires

Questionnaires for data collection from participating women were developed by the research team. A number of questions were taken from the Migrant Friendly Maternity Care Questionnaire (MFMCQ),[43] in some cases slightly modified to fit the purpose of our study, as is the intention in use of the MFMCQ, and for use both in

pregnancy and after birth, including occasional changes of tense from the original. Selected questions were also included from the Cambridge Worry Scale (11 single-item questions on worries about the upcoming labour and birth), which is available in English and Swedish,[44] and the 10-item Edinburgh Postnatal Depression Scale (EPDS) which is available in English, Swedish and Somali.[45] The questionnaires were developed in English by the research team and translated to Swedish by one of the bilingual team members and then cross-checked by another bilingual researcher in the team. Thereafter, the questionnaire was translated into Somali by a professional Somali translator, fluent in English, Swedish and Somali and finally cross-checked by a Somali-speaking research assistant. At each step of the process, the questionnaire was pilot tested with Somali-born women for relevance and understanding with adjustments made accordingly. During piloting, it was clear that some of the items and response alternatives in the EPDS were difficult for women. Although the scale is translated into Somali, no validation study has to date been published. As data are collected in face-to-face or telephone interviews conducted by the bilingual research assistant, a decision was made to include the EPDS as translated, with the research assistant able to explain the meaning of the questions in plain language if needed. Alongside the current study, we have commenced a process for validating the EPDS Somali translation, involving think-aloud interviews[46] with Somali-born women to contribute to evidence about the use of the EPDS in Somali.

### Outcome measures

Primary outcomes are women's overall ratings of antenatal care and emotional well-being. Ratings of care are assessed by the core question 'When thinking about your overall experience of antenatal care—in general, have you been happy with the care that you have received?' with response alternatives always, mostly, sometimes, rarely and never. The question is modified from MFMCQ[43] and more detailed questions about specific aspects of care are also included. Emotional well-being is assessed with EPDS, completed in English, Swedish or Somali as appropriate.[45]

Secondary outcomes are adequate number of visits measured by the Adequacy of Prenatal Care Utilization Index, that is, expected number of visits in relation to actual number adjusted for gestational week at booking visit and delivery,[47] social support during pregnancy using slightly modified questions from the Pregnancy Risk Assessment Monitoring System[48] and worries about the upcoming birth using questions from the Cambridge Worry Scale.[44] Knowledge about danger signs and where to seek healthcare if particular symptoms arise are assessed by questions with multiple choice responses, such as 'Now at the end of your pregnancy, where would you turn if you had bleeding from the vagina? When would you seek care?'. From the patient record, the following are retrieved: estimated time spent with the antenatal

care midwife, gestational age at the first ANC visit, lowest measurements of haemoglobin and S-ferritin, weight gain, attendance at parent education, mode of birth, interventions (induction of labour, pain relief including epidural, oxytocin), blood loss, diagnoses (mother) and gestational age, birth weight, Apgar score, umbilical cord pH and the need for neonatal intensive care (infant).

## Preparing for implementation
### Awareness raising
To create awareness and interest in the project and subsequently in the new model of care, Somali and other community networks are involved in different ways. Information is disseminated by participants in the reference group, by researchers and research assistants one to one and in small groups gathering Somali-born women. A brief pamphlet in Swedish and Somali has been produced and distributed among women, Somali community associations, open playgroups and activity centres and as a poster in the waiting rooms at participating ANC clinics.

### Training of midwives and interpreters
Areas to be covered in a preparatory training workshop for midwives were identified through findings in the FGDs, in dialogue with the midwives at both sites and in reference group meetings. The research team then tailored a 1 ½ day workshop for all midwives at both sites, held prior to intervention commencement. In response to the needs expressed, the main focus was group processes including PCC[37] and MI in groups.[39] A 2-day follow-up session is planned during the intervention period to enable feedback on implementation and address any issues arising.

The interpreters' preparation included 2 hours of information regarding the structure of the ANC care system, introduction to the Hooyo Project and the gANC model emphasising dialogue and MI principles, participation in meetings with involved midwives and receiving a written project manual (described below).

### Development of material for midwives leading group sessions
A manual comprising information and suggestions for the content and structure of group sessions has been developed, which the midwives are free to use and adjust according to the needs in the groups. An overview of the availability of evidence-based resources related to pregnancy and birth led us to recommend the use of existing resources where available, as they are free of charge and often in different languages. This also adds to the sustainability and cost-effectiveness of the intervention and makes it easier to duplicate. A study-specific homepage was created, comprising the same information and support as in the written manual.

## Phase III: design and methods for the historically controlled evaluation of the intervention
### Participants and recruitment
At both sites, primiparous and multiparous Somali-born women are recruited to the study. Recruitment and retention of participants to clinical studies is a well-documented challenge in research across languages and cultures.[49] We therefore thoroughly planned how to optimise the recruitment and information processes in relation to eligible women and adjusted processes to local prerequisites at both sites. The Somali-speaking research assistant is based at site 1 where most needed and where the majority of the recruitment is performed. In site 2, team member MA is involved with recruitment, information and data collection, supported by the Somali-speaking research assistant form site 1. All pregnant Somali-born women attending the two ANC clinics receive brief information about the study from the midwives during their first visit, and if they agree to receive further information, they are then contacted by the research assistant or MA after the first ANC visit, either by telephone or face to face for further oral and written information in Somali, English or Swedish, as needed.

Inclusion criteria are being born in Somalia and <25 weeks of gestation. Exclusion criterion is a health condition preventing participation in group antenatal care, that is, severe mental health condition or if the pregnancy care were to be transferred to a specialist obstetric clinic. During the first 18 months of the study, recruited women were included in the control group receiving standard individual care, and in the following 18 months, women are included in the intervention group. All women are informed verbally and orally about the purpose of the study and that participation is voluntary. Women included in the control group were informed that their perspectives about their care would contribute knowledge valuable for a proposed intervention study aimed at improving antenatal care for migrant women.

## Data collection for the intervention
### Questionnaires
Questionnaire data are collected at three time points (Q1, Q2, Q3). The baseline measurement (Q1) is conducted after recruitment, that is, before gestational week 25. Women are contacted for a face-to-face or telephone structured interview in Somali, Swedish or English at a place of their choice. Q1 includes questions on background (parity, education, income, occupation), migration (time of residence, reason for migration, migration status, language) and social support, psychological and physical well-being, knowledge of danger signs and the health system and expectations about the upcoming birth. Data for Q2 are collected in late pregnancy, after gestational week 35, in the same manner. Q2 includes questions on experiences of care and information during the ongoing pregnancy, knowledge of the health system, pregnancy and birth, social support and psychological and physical well-being and expectations for the upcoming birth. Q3 is administered 2 months postpartum and includes questions regarding the overall experience of pregnancy, ANC, birth and psychological and physical well-being. After each interview, a small gift is offered as a token of appreciation for women's participation.

## Medical records

Data for secondary outcome measurement will be collected from patient medical records, for all consenting participants. This includes data related to pregnancy, labour and birth and maternal and infant outcomes.

## Sample size issues

Our main purpose is to develop and test the feasibility of the intervention and study processes and in so doing, also provide information to inform sample size calculations for possible future randomised controlled trials. Nevertheless, based on available birth rate data, we estimate being able to include 100 women in the control group and 100 women in the intervention group during the study period. We estimate that the study will have the power to detect a clinically relevant increase in women's overall satisfaction with antenatal care from an expected 65% among Somali women receiving individual care to 82% in those receiving gANC (approximating rates for Swedish speaking women in a national population based study[50] with 80% power and an alpha of 20%, with 70 women in each group). To have similar power to detect differences in mean scores on the Edinburgh Postnatal Depression Scale (a hypothesised reduction from a mean of 8.0 in the control group— similar to that found in studies of migrant women, to 6.0 in the gANC group— similar to that found in Swedish population-based studies,[51] 63 women are required in each group. Allowing for 20% loss to follow-up at the time of data collection with women 2 months postpartum, 174 women will need to be recruited.

Differences between groups will be described as ORs and 95% CIs after estimation by means of logistic regression analyses. Comparison of means will be undertaken using t-tests where data are normally distributed or medians compared using Mann-Whitney U tests if not.

## Phase IV: design of the process evaluation

A process evaluation using a mixed methods approach will be conducted throughout the study.[29] The process evaluation will address whether the intervention was carried out according to plan, what was achieved, if and what adaptations were needed, who was reached by the intervention, any unexpected events and will also aim to understand the mechanisms of impact and what mediated these mechanisms. The process evaluation will also address contextual factors.[30]

## Process evaluation data collection and measures

Data collection for the process evaluation[30] will be nested in the different phases, supported by different measurement tools and activities. Moore's process evaluation framework[30] will guide the data collection, analysis and interpretation. The framework emphasises the impact of, and relationship between the intervention, implementation, mechanisms and context, and a topic guide and analysis will focus on what is achieved and how (fidelity, dose, adaptations, reach). Focus for the mechanisms of impact will be the participants' interactions with the intervention and potential mediating factors.

A session checklist for completion by the midwife after each group session will cover topics discussed, resources used, number of participating women and support people, group dialogue and the midwife's brief view of each session. The checklist was developed by the research team, discussed with the reference group and, after adjustments, presented to all included midwives for further modifications, which were addressed by the researchers before intervention start.

Participant observations during the group sessions will contribute data describing the delivery of gANC and the mechanisms of impact by describing group dynamics, dialogue and active participation by women in group sessions. A protocol for observations during group sessions developed by the research team, focuses on group mechanisms, dialogue and interaction, based on PPC and MI principles. This will provide the guide for participant observations by one or two members of the research team during a number of randomly selected group sessions.

Participant experiences of group antenatal care: Questionnaire data for all participating women and qualitative data collected from women, midwives, partners, interpreters and heads of departments will describe responses to and interactions with the intervention and their views regarding the feasibility of the intervention and mechanisms of impact. Women's experiences of the content, structure and feasibility of gANC will be assessed in the follow-up questionnaire (Q3) at 2 months postpartum. Additionally, the questionnaires will identify a sample of women with a diversity of experiences to be recruited for more in-depth individual interviews. Midwives' experiences of the same, and their perceptions regarding mechanisms of impact will be collected through qualitative FGDs or interviews as appropriate. Partners' and interpreters' views will be collected through qualitative individual interviews on completion of participation. FGD and interview data will be collected using a topic guide with open-ended questions developed by the research team and analysed using deductive content analysis according to Elo and Kyngas.[52] Field notes taken by the researchers in dialogue with research assistants, involved midwives and heads of departments will assist in describing the choices and decisions made during the implementation process.

## Patient and public involvement

The initial literature review informing the research questions included studies focusing on immigrant women's experiences and views of maternity care postmigration, including the views of Somali-born women. Initial focus group discussions for the study included Somali women and men with recent experiences of antenatal care, and these informed the development of the intervention. As described above, a study reference group involves Somali community members in all stages of the project, and a bicultural research assistant

is involved in recruiting women to the study. All participants will receive a summary of the results in their preferred language, and seminars will be held with stakeholder groups.

## DISCUSSION

The Hooyo study is an attempt to address some of the health and care disparities in the Swedish antenatal care system. The development and feasibility design of the study,[29] together with the nested process evaluation,[30] will contribute valuable information for future randomised controlled studies, as well as in the design of antenatal care interventions focused on reducing inequalities in health between immigrants and non-immigrants. The study will provide guidance on the acceptability of this model of care among Somali-born women, their partners and midwives in Sweden and on recruitment and data collection from responders who are often excluded from research due to communication barriers or marginalisation. The study will also indicate whether gANC has a positive impact on Somali women's experiences of antenatal care and might therefore be appropriate for other migrant groups.

For the midwives, the intervention provides a platform for reflection and dialogue about improving care for migrant women. This is vital, since care provision is always an interplay between caregivers and care receivers. The early inclusion of midwives' and participants' views provided space to shape the planned intervention in relevant ways in terms of content, structure and underlying core principles.[32] By emphasising dialogue, a person-centred approach and the active participation of women, partners and midwives throughout development and implementation of the intervention, we hope to develop a relevant, pragmatic and acceptable model of care for this group of immigrant women, and one that might replicate well to other settings and groups with minor adjustments. The challenge is to develop and implement an acceptable model of care addressing diverse needs of the care-receiving group and suited to local conditions, while ensuring that medical and public health guidelines are fulfilled. The findings of the study will determine if this can be achieved.

It could be argued that the study should include more antenatal care sites and a range of different immigrant groups. Funding limitations meant restricting the study to two sites and focusing on only one immigrant group, something that may limit the study's generalisability. On the other hand, it is expected that many of the strategies employed to develop and test group antenatal care in this study, and the lessons learnt in doing so, will have future relevance in improving antenatal care for immigrant women more generally.

### Ethics and dissemination

All participants are offered information about the study, informed about the voluntary nature of participation in detail and give their written consent prior to enrolment.

The findings of the study will be disseminated at relevant national and international conferences and through publications in peer-reviewed journals. Seminars involving local stakeholders in the communities, county council representatives and practitioners will further provide a platform for dissemination and reflections about lessons learnt at the end of the project. The findings of the study will also be presented via the Hooyo Project website https://ki.se/kbh/modrahalsovard-for-utlandsfodda-kvinnor-hooyo-projektet

**Author affiliations**
[1]School of Education, Health and Social sciences, Dalarna University, Falun, Sweden
[2]Department of Women's and Children's Health, Karolinska Institute, Stockholm, Sweden
[3]Mother and Child Health Research, La Trobe University, Melbourne, Victoria, Australia
[4]Womens and Childrens Health, Uppsala University, Uppsala, Sweden
[5]Antenatal Care Clinic, Domnarvet, Borlänge, Sweden
[6]Antenatal Care Clinic, Spånga-Tensta, Sweden
[7]Centre for Clinical Research Dalarna-Uppsala University, Falun, Sweden
[8]Faculty of Health and Social Sciences, Western Norway University of Applied Sciences, Bergen, Norway

**Acknowledgements** The authors would like to thank the midwives, members of the Hooyo reference group and all the Somali-born women and men who have contributed in the preparation phase of the study.

**Contributors** ES, RS, EA and BE initiated the study. UB, ES, RS, EA MA, BE, AA, FHA, KT, YL, MI and AA-B contributed to planning and design. UB, MA, ES, RS, EA AA and FH developed questionnaires and topic guides. UB drafted the manuscript, MA, ES, RS, UB, EA, BE, AA, FHA, KT, YL, MI and AB revised it. All authors read and approved the final manuscript.

**Funding** This study has received funding from the Swedish Research Council (grant number 2015-02470), Forte (grant number 2016-00957) and the Doctoral School in Health Care Sciences, Karolinska Institutet (grant number 2-144/2016).

**Disclaimer** The funding bodies have played no role in the design of the study, nor in data collection, analysis and interpretation of the data or in writing the manuscript.

**Competing interests** None declared.

**Patient consent for publication** Not required.

**Ethics approval** The study is approved by the Regional Ethical Review Board, Stockholm, Sweden, 2015-12-04, Dnr 2015/1703-31/1.

**Provenance and peer review** Not commissioned; externally peer reviewed.

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
