## [Reviewer comments · BMJ Open]

ARTICLE DETAILS

TITLE (PROVISIONAL)	Rationale, development and feasibility of group antenatal care for immigrant women in Sweden: a study protocol for the Hooyo-project
AUTHORS	Byrskog, Ulrika; Ahrne, Malin; Small, Rhonda; Andersson, Ewa; Essen, Birgitta; Adan, Aisha; Ahmed, Fardosa; Tesser, Karin; Lidén, Yvonne; Israelsson, Monika; Åhman-Berndtsson, Anna; Schytt, Erica

VERSION 1 - REVIEW

REVIEWER	Alison Craswell University of the Sunshine Coast, Australia
REVIEW RETURNED	01-Apr-2019

GENERAL COMMENTS	Line 44 of abstract, ethics and dissemination 'achieve' should be 'receive'? Again pg 18 line 30 - use of the word achieve suggests they only get a summary if they complete participation therefore being unethical if a parent wishes to withdraw. p8 Line 13 Design - regardless of context, 'subjects' is not respectful language in midwifery - please change to mothers, expectant parents or at least participants. The use of 'Phases' in the design suggests the paper is reporting results of the study - suggest changing this to remove phases from subheadings to avoid this. p12 Line 9 The intervention: Group Antenatal Care - suggest adding here reference to the model you base the group care on.
---

REVIEWER	Octavia Wiseman City, University of London UK I am a research fellow on the REACH RCT of a model of group antenatal care (Pregnancy Circles) in the UK.
REVIEW RETURNED	14-Apr-2019

GENERAL COMMENTS	This article describes the design process for an implementation study will add to the evidence on the acceptability of group antenatal care amongst vulnerable groups, in particular comparing single-ethnicity groups with mixed groups. There is currently considerable interest internationally in the implementation of group antenatal care as a way of improving women's experiences and
--

outcomes of antenatal care. Overall this is a carefully thought-out protocol whose strength lies in the extensive engagement with a vulnerable group of service users during the design period. Minor amendments are needed, but overall the publication of this protocol will inform the development of other studies and avoid duplication as well as ensuring rigour of the study itself.

Detailed suggestions are included below:

ABSTRACT:

Some attention to the English would help, for example Line 23 could read 'A feasibility study incorporating' rather than 'with'; Line 25 could read 'Phase II includes the development' and line 43 'voluntariness' might better be described as participants having the right to decline or withdraw without their care being affected.

Line 23 – you use the term 'historically controlled evaluation' which is confusing but this term is not used in the body of the article and clarification is needed.

Line 32 – group antenatal care is a complex intervention, not a theory – please clarify what you mean

Methods & Analysis – You present the four phases without clarifying that this article focuses on phase I & II and presents the protocol for Phases III & IV. Also, no timeline for Phase III & IV, and no dates showing when phase I & II were carried out are included here or in the article. This is important so that we understand if this is a planned or on-going study. BMJ Open cannot publish a protocol for a completed study.

ARTICLE

Page 4 – top line – surely alternative theories might be that Somali women are non-compliant, have more complex medical histories or receive a lower quality of care due to conscious or unconscious bias? Otherwise the background to the issues is well described.

Page 4 – last paragraph. There are different models of group antenatal care, but traditionally (for example the Centering model) integrated short clinical checks (3-5 minutes) within the group time, not before and after. I also suggest mentioning two of the other cornerstones of this model of care: teaching women to self-check their blood pressure and urine and providing continuity of carer (it would be interesting to know whether continuity is offered under traditional care in Sweden. You can discuss any adaptations you have made to the model, and the rationale for these, under Phase II.

Page 5 – line 30. In your evaluation of mechanisms it will be important to explore to what extent improved language support impacts on outcomes and experience, separate from the intervention itself.

Page 5 – The aims are confusing as they do not follow the four-phases outlined in the abstract. It is important to distinguish the aims of the overall study, and the aims of the pre-feasibility study presented in this article.

Page 6 – line 9. I do not feel that these references, which look at evaluating complex evaluations, add information about what your approach will be.

Page 6 – 'Design' – please clarify the timeline.

Page 9, Line 17 – please clarify what question your literature review was asking as this is not clear here. Were other interventions considered?

Page 9, Line 57. Please clarify why you chose to offer group antenatal care only to Somali women on one site, when your findings engaging service users (page 8) specifically said that 'Instead of focusing on specific cultural or ethnic aspects of care the results indicated a need to maintain focus on diversity'.

Page 9, Line 60 – Was there any finding in the FGD study about the importance of continuity of midwives?

Page 10, line 12 – Clarification is needed on how group antenatal care model has been defined/modified for the purposes of this study. What is the rationale for including 6-8 women (line 17)? How will partners be included and how this might affect the women who choose to participate? Also (line 33), please clarify how 8 women could receive 15-minute individual appointments 'before and after' a 1-hour group session, in a 1.5-hour slot. How will self-checking (if this is included in your model) also fit in to this? How the 'individual' appointment are carried out (i.e. in the group space on a mat on the floor, or in a separate room, behind a screen etc?) should also be clarified – as a complex intervention, each element should be clearly described for reproducibility.

Page 10, Line 46 – The process and issues of translating into Somali are well described, but clarification is needed on how these validated scales have been modified.

Page 11, line 10 – Instead of 'Parallel' this could read 'Alongside the feasibility study, a validation process...'. This is a useful addition to this study. Can you clarify how research assistants 'will be ready to explain in plain language' at each stage (for example, postnatal questionnaires),

Page 11, line 22 – I would suggest that the tense under 'Awareness raising' should either be the past (has this occurred?) or in the future (are you describing what will happen during the implementation)?

Page 12, line 11 I suggest this is titled 'Design of the feasibility study: a Historically Controlled Evaluation' (or 'Protocol for the feasibility study' or 'Protocol for the implementation'). The language used to describe the study should be the same as in the abstract, and it should be clear that this part of the article is describing your design, not work already undertaken (even if the work is currently in process). As such the tense should be in the future ('will be' rather than 'are'). I also think that a short clarification of why you have chosen a historically controlled evaluation as your study design (when you are not powered to actually compare outcomes). The timeline and sample sizes should also be clarified – as this is a descriptive feasibility study it is not expected that you will be powered to compare outcomes in the two groups so the purpose of the data you are collecting should be clarified: will the comparison data contribute to the design of future experimental studies or is the focus to unpick/better understand the mechanism of this complex intervention within this population?

Page 12, line 40 – What will the women being recruited into the study during the first 18 months (the control group) be told the purpose of the study is? Could consent processes be clarified?

Page 12, line 49 – The details of these outcome measures might sit more comfortably in page 10 in the 'development of the evaluation tools' section?

Page 13, line 51 – could you clarify what the 'small gift' will be? There is a lot of literature on the role incentives play in recruitment, especially with potentially poor participants.

Page 14, line 22 – Again, this could be titled 'Design of the process evaluation' or 'Protocol for the process evaluation' and, again, the tense should be in the future.

Page 14, line 30 – Could you clarify here what analytical frameworks and theory you will be using and how you plan to analyse the qualitative data? You mention Moore's process

	evaluation framework on page 15, but more detail up front to frame your approach to the process evaluation would be helpful. Page 15, line 5 – Could you clarify why staff, partners and interpreters will be interviewed or be part of focus groups with open-ended questions, but women will apparently only have questionnaires administered? Will these include open questions? Are heads of department to be interviewed, or are their views only to be included via field notes? How will interview participants be identified (convenience sampling?) and how will consent processes work? Page 15, line 13 – you can delete ‘with’ so it reads ‘will contribute valuable information’. Page 15, line 35 – there is discussion here about the strengths of the study, but discussion of the limitations if also needed. It will be interesting to hear how this model of care works with this community, in particular any differences between the single-ethnicity groups and the mixed groups, and the impact of different approaches to interpreting in both.
--	--

VERSION 1 – AUTHOR RESPONSE

Reviewer’s comment	Response from authors
Reviewer 1	
Line 44 of abstract, ethics and dissemination 'achieve' should be 'receive'?	Thank you, this has been corrected and the abstract has been re-written for clarity.
Again pg 18 line 30 - use of the word achieve suggests they only get a summary if they complete participation therefore being unethical if a parent wishes to withdraw.	Receive is the word we are looking for. Corrected.
p8 Line 13 Design - regardless of context, 'subjects' is not respectful language in midwifery - please change to mothers, expectant parents or at least participants.	Thank you for noticing this. It is now changed to women.
The use of 'Phases' in the design suggests the paper is reporting results of the study - suggest changing this to remove phases from subheadings to avoid this.	We do not really agree that describing the study in phases suggests that the paper is reporting study results. We have found it useful to describe the study as occurring in phases, for example when we discuss the project with stakeholders and others. Since the overall study includes many different actions and steps, we prefer to keep the term “Phases”, as we believe this provides readers a clearer overview of the steps taken in preparing for, developing, implementing and evaluating this complex intervention.
p12 Line 9 The intervention: Group Antenatal Care - suggest adding here reference to the model you base the group care on.	We have drawn on the group model previously implemented in Sweden and described in Andersson et al, 2017; and developed this further in light of feedback in Phase 1 so that it is suited to the needs of Somali women and the midwives

	at the study sites. This is now included in the paper on p 11 and 13.
Reviewer 2	
'This article describes the design process for an implementation study will add to the evidence on the acceptability of group antenatal care amongst vulnerable groups, in particular comparing single ethnicity groups with mixed groups. There is currently considerable interest internationally in the implementation of group antenatal care as a way of improving women's experiences and outcomes of antenatal care. Overall this is a carefully thought-out protocol whose strength lies in the extensive engagement with a vulnerable group of service users during the design period. Minor amendments are needed, but overall the publication of this protocol will inform the development of other studies and avoid duplication as well as ensuring rigour of the study itself.	Thank you! We appreciate all the effort, time and suggestions provided.
Abstract	
Some attention to the English would help, for example Line 23 could read 'A feasibility study incorporating' rather than 'with'; Line 25 could read 'Phase II includes the development' and line 43 'voluntariness' might better be described as participants having the right to decline or withdraw without their care being affected.	We have revised the text in the abstract and we have also reviewed the English throughout the manuscript.
Line 23 – you use the term 'historically controlled evaluation' which is confusing but this term is not used in the body of the article and clarification is needed.	We have clarified "historically controlled evaluation" by reporting the time periods for the data collection from controls and from women partaking in the intervention. In the main body of the text more detailed information and explanation is provided, p 6 and p 16.
Line 32 – group antenatal care is a complex intervention, not a theory – please clarify what you mean	Thank you for noticing this! The sentence describes the theoretical frame/core values underpinning the study and mentioning group antenatal care here is not relevant. Removed.
Methods & Analysis – You present the four phases without clarifying that this article focuses on phase I & II and presents the protocol for Phases III & IV. Also, no time-line for Phase III & IV, and no dates showing when phase I & II were carried out are included here or in the article. This is important so that we understand if this is a planned or on-going study. BMJ Open cannot publish a protocol for a completed study.	The paper focuses on the design of all four phases of what is an ongoing project, where the first two phases are complete. We have clarified this briefly in the abstract and more in detail, with timelines in the main article text, p 6.

Article	
Page 4 – top line – surely alternative theories might be that Somali women are non-compliant, have more complex medical histories or receive a lower quality of care due to conscious or unconscious bias? Otherwise the background to the issues is well described.	Yes there are many potential explanations. Text is now added to more fully describe these complex possibilities, p 3-4. We prefer to use the word discrimination instead of bias for the understanding of clinical midwives and because discrimination is what migrant women talk about, in our study, and in previous studies.
Page 4 – last paragraph. There are different models of group antenatal care, but traditionally (for example the Centering model) integrated short clinical checks (3-5 minutes) within the group time, not before and after. I also suggest mentioning two of the other cornerstones of this model of care: teaching women to self-check their blood pressure and urine and providing continuity of carer (it would be interesting to know whether continuity is offered under traditional care in Sweden. You can discuss any adaptations you have made to the model, and the rationale for these, under Phase II.	We have now included text about self assessment and the clinical checks, 4-5. The group antenatal care initiatives conducted in Sweden prior to this study, and followed in the present model, include short individual appointments with the midwife before or after the group session, which differs from, for instance the Centering Pregnancy model. This is clarified in the intervention section, p 12-13. Continuity of care with the same midwife is the norm within the traditional Swedish antenatal care, which we mention on page 4. This explains our lack of emphasis on this aspect in relation to group antenatal care, as poorer experiences of care among immigrants in Sweden are unlikely to be explained by lack of continuity of the antenatal midwife, as may be a contributing factor in other settings.
Page 5 – line 30. In your evaluation of mechanisms it will be important to explore to what extent improved language support impacts on outcomes and experience, separate from the intervention itself.	True. In the questionnaires - with both women participating in the intervention and women in the control group – experiences of language support and communication aspects are central.
Page 5 – The aims are confusing as they do not follow the four-phases outlined in the abstract. It is important to distinguish the aims of the overall study, and the aims of the pre-feasibility study presented in this article.	To minimise the confusion regarding the aims we have now clarified what is the overall aim and what are the specific objectives. The different phases have been more clearly linked to the different specific objectives, p 5.
Page 6 – line 9. I do not feel that these references, which look at evaluating complex evaluations, add information about what your approach will be.	We argue that these references support and justify the use of a flexible and context-sensitive study design and examples of how this has guided the study design and process are given in the points in next sentence, p 6.
Page 6 – ‘Design’ – please clarify the timeline.	A timeline is added on page 6.
Page 9, Line 17 – please clarify what question your literature review was asking as this is not	We have added more explanatory text regarding the focus for the literature review, p 11.

clear here. Were other interventions considered?	
Page 9, Line 57. Please clarify why you chose to offer group antenatal care only to Somali women on one site, when your findings engaging service users (page 8) specifically said that 'Instead of focusing on specific cultural or ethnic aspects of care the results indicated a need to maintain focus on diversity'.	This is related to the language issue. Since a large number of Somali migrants had settled in the municipality of site 1 during recent years and do not speak Swedish well, an arena for women to speak in their native language was needed with support for interpreting to improve dialogue with midwives – this was a parallel finding in the focus group discussions. By providing this language supported forum, it is hoped that we will overcome the tendency to resort to stereotypes when communication is inadequate.
Page 9, Line 60 – Was there any finding in the FGD study about the importance of continuity of midwives?	In Swedish antenatal care, continuity of midwife care is standard. This might be the reason this aspect was not brought up specifically in the FGDs. We have added information about continuity of care in the Swedish ANC system in the background text.
Page 10, line 12 – Clarification is needed on how group antenatal care model has been defined/modified for the purposes of this study. What is the rationale for including 6-8 women (line 17)? How will partners be included and how this might affect the women who choose to participate? Also (line 33), please clarify how 8 women could receive 15-minute individual appointments 'before and after' a 1-hour group session, in a 1.5-hour slot. How will self-checking (if this is included in your model) also fit in to this? How the 'individual' appointments are carried out (i.e. in the group space on a mat on the floor, or in a separate room, behind a screen etc?) should also be clarified – as a complex intervention, each element should be clearly described for reproducibility.	We have added more information in the text to clarify the model, p 12-13. In Swedish ANC, partners are encouraged to participate during the ANC visits and this is therefore the point of departure for this study. Since women may have different views of this, each group will come to a consensus about whether partners will be invited to participate or not. Self-checking is not a part of this model; since the included midwives considered this task to be a part of their professional responsibility. Regarding the 15 minute individual appointment, this check-up will be with the midwife responsible for each woman (p 12), who may or may not be one of the midwives leading the group; this will make it possible to do several check-ups at the same time and these will be carried out in separate rooms nearby.
Page 10, Line 46 – The process and issues of translating into Somali are well described, but clarification is needed on how these validated scales have been modified.	We have modified the text for greater clarity as follows (p 13): A number of questions were taken from the Migrant Friendly Maternity Care Questionnaire (MFMCQ)(41), in some cases slightly modified to fit the purpose of our study and for use in pregnancy and after birth, including occasional changes of tense from the original. Selected questions were also included from the Cambridge Worry Scale (11 single-item questions on worries about the upcoming labour and birth), which is available in English and Swedish(42), and the ten-item Edinburgh Postnatal Depression Scale

	(EPDS) available in English, Swedish and Somali(43). Note: The EPDS is included in full and the translations have not been modified.
Page 11, line 10 – Instead of ‘Parallel’ this could read ‘Alongside the feasibility study, a validation process...’. This is a useful addition to this study. Can you clarify how research assistants ‘will be ready to explain in plain language’ at each stage (for example, postnatal questionnaires),	Thank you, changed accordingly. Since the research assistant interviews women using the structured questionnaire, she is present and able to explain the meaning of any question that a women might have trouble with. This is now clarified in the text, p 13.
Page 11, line 22 – I would suggest that the tense under ‘Awareness raising’ should either be the past (has this occurred?) or in the future (are you describing what will happen during the implementation)?	The awareness raising is a continuous process of keeping the community aware of the project at each stage; it started prior to the intervention commencing and is ongoing alongside the intervention. Therefore we prefer to keep the present tense here.
Page 12, line 11 I suggest this is titled ‘Design of the feasibility study: a Historically Controlled Evaluation’ (or ‘Protocol for the feasibility study’ or ‘Protocol for the implementation’). The language used to describe the study should be the same as in the abstract, and it should be clear that this part of the article is describing your design, not work already undertaken (even if the work is currently in process). As such the tense should be in the future (‘will be’ rather than ‘are’). I also think that a short clarification of why you have chosen a historically controlled evaluation as your study design (when you are not powered to actually compare outcomes). The timeline and sample sizes should also be clarified – as this is a descriptive feasibility study it is not expected that you will be powered to compare outcomes in the two groups so the purpose of the data you are collecting should be clarified: will the comparison data contribute to the design of future experimental studies or is the focus to unpick/better understand the mechanism of this complex intervention within this population?	Thank you, we have modified this heading to Design and methods for the historically controlled evaluation of the intervention Yes, the tense changes throughout the article. Since the different steps in the phases in this study are building on each other we were not able to write a protocol paper from the very beginning. Now in the middle of the process it is possible to describe the full protocol and this is the reason why the tense changes in the article. What has been conducted is described in the past tense, ongoing phases are in present tense and what is yet to be done in future tense. We have made a number of changes here in response to these comments. A power calculation in relation to our two primary outcomes shows that 70 women in each group will be sufficient, based on previous literature on migrant and non-migrant women’s ratings of care and EPDS mean scores, to detect clinically relevant differences. This is now clarified, page 17. The timelines have been outlined on page 6. We have clarified in the text, page 17 that the study findings can contribute to the design of future experimental studies. In the discussion, page 1 we describe our dual purpose; the contribution to future experimental studies regarding sample size and outcome measures, and to the future design of antenatal care interventions that target inequalities in health between immigrants and non-immigrants.

Page 12, line 40 – What will the women being recruited into the study during the first 18 months (the control group) be told the purpose of the study is? Could consent processes be clarified?	Women included in the control group receive information that their perspectives about present care will contribute knowledge valuable for a future intervention study aimed at improving antenatal care. This we have clarified in the text, p 16.
Page 12, line 49 – The details of these outcome measures might sit more comfortably on page 10 in the ‘development of the evaluation tools’ section?	We agree, thank you, this section has been moved.
Page 13, line 51 – could you clarify what the ‘small gift’ will be? There is a lot of literature on the role incentives play in recruitment, especially with potentially poor participants.	After completion of questionnaire 1 (Q1) participating women in both control and intervention groups are offered a t-shirt suitable for breastfeeding, after Q2 some small samples of body lotion, and upon completion of Q3 a small toy for the baby.
Page 14, line 22 – Again, this could be titled ‘Design of the process evaluation’ or ‘Protocol for the process evaluation’ and, again, the tense should be in the future.	The heading is changed to ‘Design of the process evaluation’ and we have changed the tense to future tense.
Page 14, line 30 – Could you clarify here what analytical frameworks and theory you will be using and how you plan to analyse the qualitative data? You mention Moore’s process evaluation framework on page 15, but more detail up front to frame your approach to the process evaluation would be helpful.	We have added more details in the text, p17 and 18. Moore’s framework for complex interventions will be the analytical framework used. The different components in this framework outlining implementation and mechanisms of impact will guide the data collection and analysis, supported by deductive content analysis based on Elo and Kyngas (2008).
Page 15, line 5 – Could you clarify why staff, partners and interpreters will be interviewed or be part of focus groups with open-ended questions, but women will apparently only have questionnaires administered? Will these include open questions? Are heads of department to be interviewed, or are their views only to be included via field notes? How will interview participants be identified (convenience sampling?) and how will consent processes work?	Thank you for this comment. We propose to supplement the questionnaire data for women also with qualitative interviews with selected participating women to explore their perspectives in more detail. This is now included in the text, p 18. Our research assistant will support our recruiting participants with as varied perspectives as possible based on data gathered through the questionnaires. This is to ensure we hear both positive and more critical voices in relation to the intervention. The research assistant will contact women and oral and written consent will be obtained prior to the interview. Heads of departments will also be interviewed.
Page 15, line 13 – you can delete ‘with’ so it reads ‘will contribute valuable information’.	Thank you, adjusted accordingly.
Page 15, line 35 – there is discussion here about the strengths of the study, but discussion of the limitations if also needed.	We have responded to this by adding to the text on strengths and limitations, p 19-20. The study includes two sites only and is focused on one immigrant group, largely due to funding

	limitations. This limits the generalisability of the findings.
It will be interesting to hear how this model of care works with this community, in particular any differences between the single-ethnicity groups and the mixed groups, and the impact of different approaches to interpreting in both.	Agree!

VERSION 2 – REVIEW

REVIEWER	Octavia Wiseman City, University of London UK Research Fellow on REACH Pregnancy Programme randomised controlled trial of group antenatal care in the UK
REVIEW RETURNED	03-Jun-2019

GENERAL COMMENTS	I think one more line at the end of Page 6 (after the explanation of the timeline) would be useful to explain exactly what this paper covers.
---

VERSION 2 – AUTHOR RESPONSE

Reviewers comment:

"I think one more line at the end of Page 6 (after the explanation of the timeline) would be useful to explain exactly what this paper covers".

Response by authors:

Thank you, to clarify this to the reader we have added a line at page 6:

"The paper covers methodological aspects of all four phases of the study."

In the main document this is added without the change marked-up and in the marked up manuscript with a yellow-marked change.